# Denoising Diffusion Causal Discovery

## Abstract

A common theme across multiple disciplines of science is to understand the underlying dependencies between variables from observational data. Such dependencies are often modeled as Bayesian Network (BNs), which by definition are Directed Acyclic Graphs (DAGs). Recent advancements, such as NOTEARS and DAG-GNN, have focused on formulating continuous DAG constraints and learning DAGs via continuous optimization. However, these methods often have scalability issues and face challenges when applied to real world data. In this paper, we propose Denoising Diffusion Causal Discovery (DDCD), a new learning framework that leverages Denoising Diffusion Probabilistic Models (DDPMs) for causal structural learning. Using the denoising objective, our method allows the model to explore a wider range of noise in the data and effectively captures both linear and nonlinear dependencies. It also has reduced complexity and is more suitable for inference of larger networks. To accommodate potential feedback loops in biological networks, we propose a k-hop acyclicity constraint. Additionally, we suggest using fixed-size bootstrap sampling to ensure similar training performance across varying dataset sizes. Our experiments on synthetic data demonstrate that DDCD achieves consistent competitive performance compared to existing methods while noticeably reducing computation time. We also show that DDCD can generate trustworthy networks from real-world datasets.

## 1 Introduction

Learning a network structure that represents underlying causal dependencies between variables in observational data has long been a crucial goal. It plays an important role in multiple disciplines, including genetics, epidemiology, and economics (Slonim, 2002; Pearl, 2009; Uhler et al., 2013; Spirtes & Zhang, 2016; Pingault et al., 2018; Glymour et al., 2019). Such networks, where nodes represent feature variables and edges capture potential relationships, are often represented as Directed Acyclic Graphs (DAGs), which allow no cycles.

There has been a great deal of prior work developing methods to address this problem. The PC algorithm (Spirtes & Glymour, 1991) is a constraint-based approach that iteratively testing conditional independence. GES (Chickering, 2002) is a score-based method that searches for the causal structure that maximizing a scoring function. LiNGAM (Shimizu et al., 2012), on the other hand, is a structural equation model (SEM)-based method. However, they scale poorly as the size of data increases; in fact, the learning inference problem has been proven to be NP-hard (Chickering et al., 2004). To address this challenge, Zheng et al. (2018) proposed a method called NOTEARS that introduces a continuous acyclicity score, which can be solved by a regular numerical optimizer.

Many others have since extended this formulation, focusing on scalability (Yu et al., 2019; Lee et al., 2019; Ng et al., 2020; Yu et al., 2021), convexity (Bello et al., 2022; Deng et al., 2023; Ng et al., 2024), sparsity control (Wei et al., 2020; Ng et al., 2020), and nonlinearity (Yu et al., 2019; Ng et al., 2019; Zheng et al., 2020; Yang et al., 2021; Shen et al., 2022; Ng et al., 2022; Kalainathan et al., 2022). Its variations also have been applied to a wide range of settings, such as time series (Pamfil et al., 2020; Sun et al., 2021; Shang et al., 2021) and gene networks (Shu et al., 2021; Agamah et al., 2022; Zhu & Slonim, 2024). There are also analyses and discussions of the application of such models to datasets with unequal variances and different data types (Reisach et al., 2021; Kaiser & Sipos, 2021; Ng et al., 2024). Methods that focus on topological ordering instead of a DAG structrual constraint have also been explored (Sanchez et al., 2022).

In this paper, we propose a set of Denoising Diffusion Causal Discovery (DDCD) models that offer significant improvements on scalability and the models' ability to capture nonlinear transformations. DDCD was inspired by the designs of Denoising Diffusion Probabilistic Models (DDPMs) (Ho et al., 2020) and Latent Diffusion Models (LDMs) (Rombach et al., 2022). In DDCD, we introduce noise progressively into the data during a forward diffusion process and then reverse the process by predicting the added noise under the constraint of the learned adjacency matrix. By replacing the least squares loss in NOTEARS with a denoising objective, DDCD allows the model to explore a broader range of noise, which enhances its ability to capture both linear and nonlinear relationships. We demonstrate the effectiveness of the proposed method on both synthetic data and large scale real-world datasets. In summary, our contributions are as follows:

- We prove the validity of the denoising objective, which paves the way to using diffusion models for causal structural learning.

- We push the boundary of structural learning on nonlinear data by showing that the nonlinear transformation function can be approximated together with the adjacency matrix.

- We introduce a k-hop acyclicity constraint, which approximates acyclicity within a k-hop neighborhood. It is a relaxation of the acyclicity constraint in NOTEARS and has improved complexity. We also applied gradient clipping to avoid gradient explosion when the graph is large. We discuss when acyclicity is helpful and when it may not be.

- We propose a fixed-size bootstrap sampling technique so the learning process proceeds similarly for data sets of different sizes.

## 2 BACKGROUND AND RELATED WORK

### 2.1 PROBLEM STATEMENT

Given a dataset $\boldsymbol{X} \in \mathbb{R}^{n \times d}$, where $n$ is the number of samples and $d$ is the number of feature variables, the objective is to learn a meaningful dependency graph $\mathcal{G}$ represented by the weighted adjacency matrix $\boldsymbol{W} \in \mathbb{R}^{d \times d}$. Such a graph is often defined as a Bayesian Network (BN), which by definition is a Directed Acyclic Graph (DAG), where there are no cycles or self loops.

### 2.2 STRUCTURAL EQUATION MODELS (SEMS)

Structural Equation Models (SEMs) (Kline, 2023) provide a framework to model variable dependencies. For a linear SEM, we simply assume that each variable is a linear combination of its parents with some noise. In its matrix multiplication form, we have

$$\boldsymbol{X} = \boldsymbol{X}\boldsymbol{W} + \boldsymbol{E}, \tag{1}$$

where $\boldsymbol{E} \in \mathbb{R}^{n \times d}$ captures the error terms. Based on this assumption, many existing SEMs aim to estimate matrix $\boldsymbol{W}$ such that the reconstruction error is minimized (van de Geer & Bühlmann, 2013; Loh & Bühlmann, 2014; Zheng et al., 2018). Since the adjacency matrix is often sparse, many methods choose to add either L1 or L2 regularization on $\boldsymbol{W}$ to encourage sparsity (Vowels et al., 2022). In this case, we have the following training objective,

$$\min_{W} \frac{1}{2n} \|\boldsymbol{X} - \boldsymbol{X}\boldsymbol{W}\|_F^2 + \lambda_1 \|\boldsymbol{W}\|_1 + \lambda_2 \|\boldsymbol{W}\|_2 \tag{2}$$

To extend the use of SEMs to real world applications, where nonlinearity is common, we rewrite Equation 1 in the following form, where $f$ is a nonlinear transformation function:

$$\boldsymbol{X} = f(\boldsymbol{X}; \boldsymbol{W}) + \boldsymbol{E} \tag{3}$$

In practice, Equation 3 allows too much freedom of model formulation. Therefore, people often use the following equation to describe a nonlinear SEM, where $f_1$ is the nonlinear transformation function for $\boldsymbol{X}$ before it meets $\boldsymbol{W}$ and $f_2$ is the nonlinear transformation function for the product of graph propagation (Yu et al., 2019; Ng et al., 2019).

$$\boldsymbol{X} = f_2(f_1(\boldsymbol{X})\boldsymbol{W}) + \boldsymbol{E} \tag{4}$$

## 2.3 CONTINUOUS DAG CONSTRAINT

Traditional network inference approaches often rely on combinatorial optimization, which becomes computationally infeasible for large graphs. To address this issue, Zheng et al. (2018) proposed a method called NOTEARS that introduced a continuous score characterizing graph acyclicity:

$$h(\boldsymbol{W}) = \text{tr}(e^{\boldsymbol{W} \circ \boldsymbol{W}}) - d, \tag{5}$$

where $\circ$ is the Hadamard product, $e^{\boldsymbol{W}}$ is the matrix exponential of $\boldsymbol{W}$, and $\text{tr}()$ is the trace of a matrix. Essentially, matrix $\boldsymbol{W}$ is a DAG if and only if $h(\boldsymbol{W}) = 0$. Since the function $h(\boldsymbol{W})$ has a simple and smooth gradient function, it can be used in many gradient-based continuous optimization algorithms. In NOTEARS, the DAG $\boldsymbol{W}$ is learned by optimizing a modification of Equation 2 (with only the L1 loss) while keeping $h(\boldsymbol{W})$ near zero with an augmented Lagrangian method using a L-BFGS optimizer. In terms of complexity, since the score function $h(\boldsymbol{W})$ requires the matrix exponential, the runtime of NOTEARS is at least $\mathcal{O}(d^3)$.

## 2.4 DENOISING DIFFUSION PROBABILISTIC MODELS

Denoising Diffusion Probabilistic Models (DDPMs) are a class of generative model that shows strong performance in modeling complex data distributions (Ho et al., 2020). A typical DDPM starts from a non-parameterized forward diffusion process. Given an unperturbed input $\boldsymbol{x}_0$, the forward process aims to generate a series of noisy samples $\boldsymbol{x}_0, \boldsymbol{x}_1, ..., \boldsymbol{x}_T$ over $T$ steps, where $\boldsymbol{x}_T$ usually stands for pure noise. In each step, a small amount Gaussian noise is gradually introduced following a diffusion schedule $\beta$ as shown in Equation 6.

$$\boldsymbol{x}_t = \sqrt{1 - \beta_t}\boldsymbol{x}_{t-1} + \sqrt{\beta_t}\boldsymbol{z}_{t-1} \tag{6}$$

Here, $\boldsymbol{z}_{t-1} \sim \mathcal{N}(0, 1)$ so Equation 6 is essentially trying to reduce the means to $\boldsymbol{0}$ while increasing the variances to $\boldsymbol{1}$. With the reparameterization trick, it can be rewritten into Equation 7 as follows,

$$\boldsymbol{x}_t = \sqrt{\overline{\alpha_t}}\boldsymbol{x}_0 + \sqrt{1 - \overline{\alpha_t}}\boldsymbol{z}, \tag{7}$$

where $\overline{\alpha_t} = \prod_{i=0}^{t}(1 - \beta_t)$ and $\boldsymbol{z} \sim \mathcal{N}(0, 1)$, so the noisy data $x_t$ can be generated in one step.

The actual modeling piece of DDPM is the reverse model, which is trained to predict the added noise $\boldsymbol{z}$ and to denoise the data. The choice of model for the reverse process depends on the input data. Recent studies have suggested similarity between diffusion models and a generalized form of variational autoencoder (VAE) (Kingma, 2013) with infinite latent spaces (Luo, 2022).

## 3 METHODS

Inspired by the denoising diffusion framework in DDPMs, here we propose an alternative training objective to learn the adjacency matrix $\boldsymbol{W}$ for a SEM. In this proposal, we will augment each sample in the same way as the forward diffusion process in Equation 7. Then, we will optimize a reverse model with the parameterized $\boldsymbol{W}$ to predict the added noise under the constraint of $\boldsymbol{W}$. We start with the assumptions for linear SEMs and then move to nonlinear cases.

### 3.1 DENOISING DIFFUSION MODELS FOR LINEAR SEMS

For linear SEMs, the reverse model is trying to minimize the following denoising objective:

$$\min_{W} \frac{1}{2n}\|(\boldsymbol{X_t} - \boldsymbol{X_t}\boldsymbol{W}) - \text{diag}(\sqrt{1 - \overline{\alpha_t}})\boldsymbol{Z}(\boldsymbol{I} - \boldsymbol{W})\|_F^2 + \lambda_1\|\boldsymbol{W}\|_1 + \lambda_2\|\boldsymbol{W}\|_2, \tag{8}$$

where $\boldsymbol{t}$ is a vector of diffusion time steps for all samples in $\boldsymbol{X}$, $\boldsymbol{X_t}$ is the perturbed observational data $\boldsymbol{X}$, and $\text{diag}(\boldsymbol{v})$ is the diagonal operator that converts vector $\boldsymbol{v}$ to a diagonal matrix.

**Theorem 1.** *For linear SEMs, the objective functions in Equation 8 and Equation 2 are equivalent.*

*Proof.* Consider the case when each sample in $\boldsymbol{X}$ is perturbed using the forward diffusion process in 7. The perturbed observational data could be written as in Equation 9. Here we use $\boldsymbol{t}$ to denote

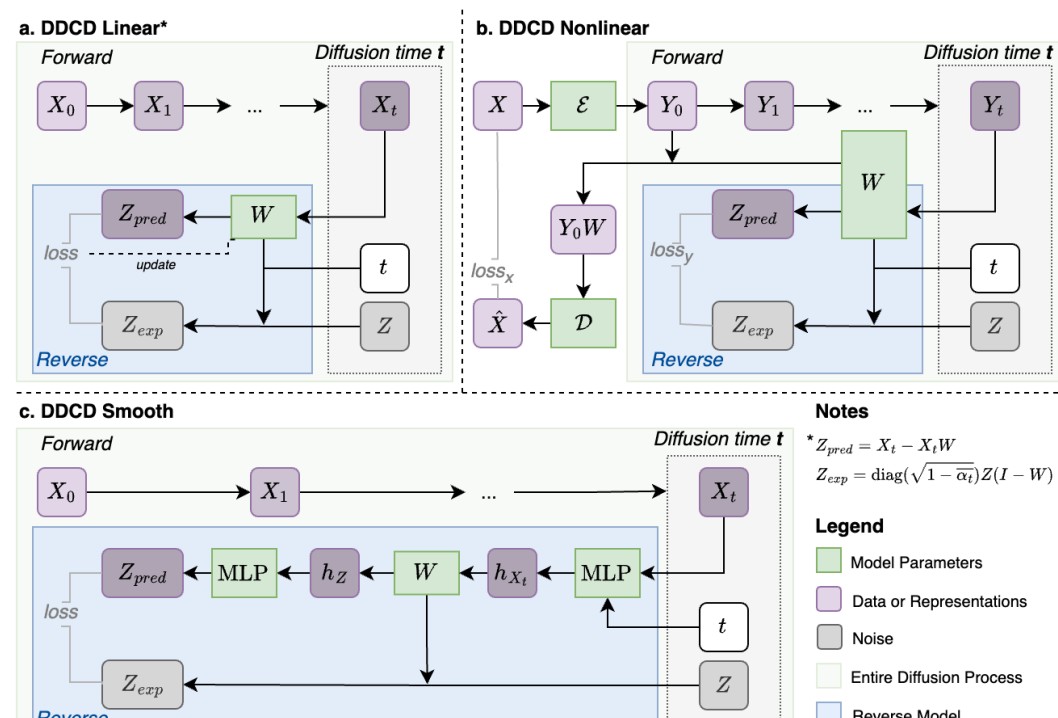

Figure 1: Model architectures of proposed models in this paper.

the different diffusion time steps for all samples in $\boldsymbol{X}$ and the diag() operator scales each row of $\boldsymbol{X_0}$ and $\boldsymbol{Z}$ accordingly based on the diffusion schedule.

$$\boldsymbol{X_t} = \mathrm{diag}(\sqrt{\overline{\alpha_t}})\boldsymbol{X_0} + \mathrm{diag}(\sqrt{1 - \overline{\alpha_t}})\boldsymbol{Z}, \tag{9}$$

With Equations 9 and 1, we can easily develop Equations 10-12

$$\boldsymbol{X_t}\boldsymbol{W} = \mathrm{diag}(\sqrt{\overline{\alpha_t}})\boldsymbol{X_0}\boldsymbol{W} + \mathrm{diag}(\sqrt{1 - \overline{\alpha_t}})\boldsymbol{Z}\boldsymbol{W}, \tag{10}$$

$$\boldsymbol{X_t} - \boldsymbol{X_t}\boldsymbol{W} = \mathrm{diag}(\sqrt{\overline{\alpha_t}})(\boldsymbol{X_0} - \boldsymbol{X_0}\boldsymbol{W}) + \mathrm{diag}(\sqrt{1 - \overline{\alpha_t}})(\boldsymbol{Z} - \boldsymbol{Z}\boldsymbol{W}) \tag{11}$$

$$\mathrm{diag}(\sqrt{\overline{\alpha_t}})(\boldsymbol{X_0} - \boldsymbol{X_0}\boldsymbol{W}) = (\boldsymbol{X_t} - \boldsymbol{X_t}\boldsymbol{W}) - \mathrm{diag}(\sqrt{1 - \overline{\alpha_t}})\boldsymbol{Z}(\boldsymbol{I} - \boldsymbol{W}) \tag{12}$$

Recall that in Equation 2, the main objective is to minimize $\boldsymbol{X_0} - \boldsymbol{X_0}\boldsymbol{W}$. Equation 12 shows that minimizing $\boldsymbol{X_0} - \boldsymbol{X_0}\boldsymbol{W}$ is equivalent to minimizing the right-hand side, which measures the distance between $(\boldsymbol{X_t} - \boldsymbol{X_t}\boldsymbol{W})$ and $\mathrm{diag}(\sqrt{1 - \overline{\alpha_t}})\boldsymbol{Z}(\boldsymbol{I} - \boldsymbol{W})$.

$\square$

Although we just showed that Equation 8 is equivalent to 2, in practice the denoising objective smooths out the gradients by adding a noise component that regularizes the learning process. This prevents large changes that could impair convergence, especially when the number of samples is limited. At the same time, by introducing noise, this new objective encourages the model to find solutions that generalize better by avoiding sharp minima (Keskar et al., 2016). These two characteristics can help the model to converge to an optimal solution more efficiently. To illustrate this point, we implemented a toy model called NOTEARS-Denoising with the denoising diffusion process, while everything else is the same as the original implementation of NOTEARS-Linear.

With the new objective in mind, we also propose a linear DDCD model, where the only trainable parameters are values in the adjacency matrix $\boldsymbol{W}$. The input of the model includes the perturbed $\boldsymbol{X_t}$, the diffusion variance schedule $\sqrt{1 - \overline{\alpha_t}}$, and the noise term $\boldsymbol{Z}$. All three inputs are generated

during the forward process given $X_0$. The model is trained with both L1 and L2 regularization on the adjacency matrix together with a k-hop acyclicity constraint, which will be explained in section 3.4. In terms of optimization, the model is optimized using the Adam optimizer with increasing weights on the DAG constraint. We will explain that in detail in section 3.6.

## 3.2 DENOISING DIFFUSION MODELS FOR NONLINEAR SEMS

The main challenge of applying the de-noising objective to non-linear cases is that, without the linear assumption on the transformation functions, we cannot simply separate the noise term $Z$ from the signals of $X$. That makes Theorem 1 not applicable to nonlinear SEMs. To overcome this challenge, we introduce an intermediate variable $Y$ and rewrite the nonlinear SEM in Equation 4 as,

$$Y = f_1(X) \tag{13}$$

$$X = f_2(YW) + E_1 \tag{14}$$

By assuming $YW = Y$, Eq. 13 and Eq. 14 may be viewed as the encoder and decoder of input $X$ while keeping all the dimensions of $X$ in the latent map $Y$. Therefore, if an adjacency matrix $W$ describes linear dependencies in $Y$, it could also be used to describe the dependencies in $X$. To effectively learn $Y$, we can use the linear denoising diffusion models we discussed previously.

$$Y_0 = Y_0 W + E_2 \tag{15}$$

Together, the model will be trained with two main objectives. First of all, we would like to minimize the nonlinear SEM reconstruction loss on $X$. Secondly, we would like to minimize the linear denoising diffusion loss on $Y$. The full loss function is,

$$\min_W \frac{1}{2n}(\|X - f_2(f_1(X)W)\|_F^2 + \|(Y_t - Y_t W) - \mathrm{diag}(\sqrt{1 - \overline{\alpha_t}})Z(I - W)\|_F^2) + \lambda_1\|W\|_1 + \lambda_2\|W\|_2, \tag{16}$$

From the deep learning point of view, this architecture, as shown in Figure 1b, is very similar to a Latent Diffusion Model (LDM) (Rombach et al., 2022), where noise is added to the latent representation learned by an autoencoder. In this case, we can treat $Y$ as the unobserved latent variable, and the learned nonlinear transformation functions $f_1$ and $f_2$ could be viewed as the encoder and decoder. Furthermore, the learned adjacency matrix $W$ for the SEM could be considered as a form of attention or graph neural network.

## 3.3 SCALE HANDLING WITH SMOOTHED FEATURES AND ADJACENCY MATRIX

While continuous optimization causal discovery algorithms have yielded great success on synthetic SEM datasets, recent studies (Reisach et al., 2021; Kaiser & Sipos, 2021) have shown that they often don't work that well on datasets with standardized features or on real-world datasets where the scales are unknown. The root cause, as pointed out in Kaiser & Sipos (2021), is the use of SEM equations in Equations 1 and 3. An underlying assumption of these two SEM equations is that the variables in $X$ must be on the same scale. If the scales of the features are altered differently due to data standardization, the equation will no longer hold. The DDCD Linear and DDCD Nonlinear methods we propose here have the same problem. To overcome this issue, we propose a smoothed version of DDCD. Instead of trying to learn the exact values in the adjacency matrix of the SEM, DDCD Smooth tries to learn a normalized adjacency, where the expected values of the edge weights are $\frac{1}{d}$. This normalized adjacency matrix is conceptually similar to the one used in graph convolution (Kipf & Welling, 2016). To do this, we first use an MLP with Tanh activation to normalize all features to the range of -1 to 1 regardless. An illustration of the architecture of this model is provided in Figure 1c. In appendix A.2, we also have a short proof that extends Theorem 1. Basically we show that in this case, instead of estimating the expected $Z$, we can directly estimate $Z$ when the number of nodes is large.

## 3.4 K-HOP ACYCLICITY CONSTRAINT WITH GRADIENT CLIPPING

In practice, the $\mathcal{O}(d^3)$ runtime of NOTEARS' DAG constraint and its risk of gradient explosion on larger networks is restrictive. Furthermore, many real world networks, such as gene regulatory net-

works, include cycles and feedback loops. Thus, we believe that starting with the DAG assumption may be incorrect, although we do want to prevent the network from being symmetric. Based on these considerations, by transforming the matrix exponential in NOTEARS to its power expansion form, we propose an alternative "k-hop acyclicity constraint" that only checks the acyclicity score within k hops. By keeping a running sum, we can reduce the runtime to $\mathcal{O}(k \cdot d^2)$. The exact formula of k-hop acyclicity is in Equation 17, where $\gamma$ is a scaling factor. A detailed explanation of this is provided in Appendix A.1 and we also include an analysis on the choice of $k$. In addition, we apply gradient clipping on the model parameters to prevent gradient explosion on large networks.

$$h(\boldsymbol{W}, k, \gamma) = \sum_{j=1}^{k+1} \frac{1}{j! \gamma^{2j}} \text{tr}((\gamma \boldsymbol{W} \circ \gamma \boldsymbol{W})^j) \tag{17}$$

### 3.5 Fixed-size Bootstrap Sampling

In this work, we use the fixed-sized bootstrap sampling design from RegDiffusion (Zhu & Slonim, 2024). Basically, in each training iteration, we sample a fixed size batch of samples with replacement and add different amounts of noise to the samples. By doing so, we remove the dependency on the number of samples $n$ from the algorithm's runtime and gain more similar behavior on data of different sizes.

### 3.6 Optimization

In this experiment, the models are optimized using the Adam optimizer since it has fewer restrictions. However, as shown in the NOTEARS-Denoising example, the denoising diffusion objective could be applied to the NOTEARS model directly and optimized with L-BFGS-B without any issues. Another experiment design we tested involves replacing the dual-ascending augmented Lagrangian method used in many methods, such as NOTEARS and DAG-GNN, with a simple linear multiplier using training epoch steps. Our justification for this is that our training pattern is much smoother with the fixed-size bootstrap sampling, so we can replace the automatic scaled Lagrangian multiplier with a simple linear multiplier. Another motivation is that in the case when edge weights are mostly smaller than 1 (for example, when features are normalized due to the use of neural networks), it no longer makes sense to use the quadratic term used in the augmented Lagrangian as a heavy penalty.

## 4 Experiments

Our first experiment is a comparison of results from NOTEARS-Denoising and NOTEARS-Linear. The only differences between these two models are the noise perturbation process and the denoising objective. Both models are optimized using dual-ascending augmented Lagrangian with the L-BFGS-B optimizer. Performance is evaluated on a synthetic Scale-Free graph with 20 nodes and degree 10. We use this example to demonstrate how the denoising objective can smooth out gradients and help the model converge faster.

Then, we evaluate the performance of the proposed linear and nonlinear models on synthetic and real data. The results for synthetic data are compared to a range of well-known causal inference methods: NOTEARS (Zheng et al., 2018), NOTEARS-MLP (Zheng et al., 2018), DAG-GNN (Yu et al., 2019), GOLEM (Ng et al., 2020), and GAE (Ng et al., 2019). We tested the performance of the models on various numbers of nodes, numbers of observations, graph types, and SEM types.

For the synthetic data, we pre-generated a set of Scale-Free (SF) and Erdős-Rényi (ER) random graphs with a wide range of node counts (20 - 5,000) and edge degrees (10 - 500). Edge weights can be fully positive (ranging from 0.5 to 1.5) or both negative and positive (ranging from -1 to 1). Observational data were generated following additive noise models (ANMs) with both linear and nonlinear transformations. For nonlinear transformations, we follow the examples of DAG-GNN(Yu et al., 2019) and tested the following nonlinear SEM: $\boldsymbol{x} = \boldsymbol{W}^T \cos(\boldsymbol{x} + \boldsymbol{1}) + \epsilon$. In both linear and nonlinear cases, data was generated with Gaussian noise. To assess the quality of the inferred structures, we report the True Positive Rate (TPR), False Discovery Rate (FDR), False Positive Rate (FPR), Structural Hamming Distance (SHD), and algorithm execution time. We focus on SHD as the main metric. A detailed description of these metrics is provided in the Appendix.

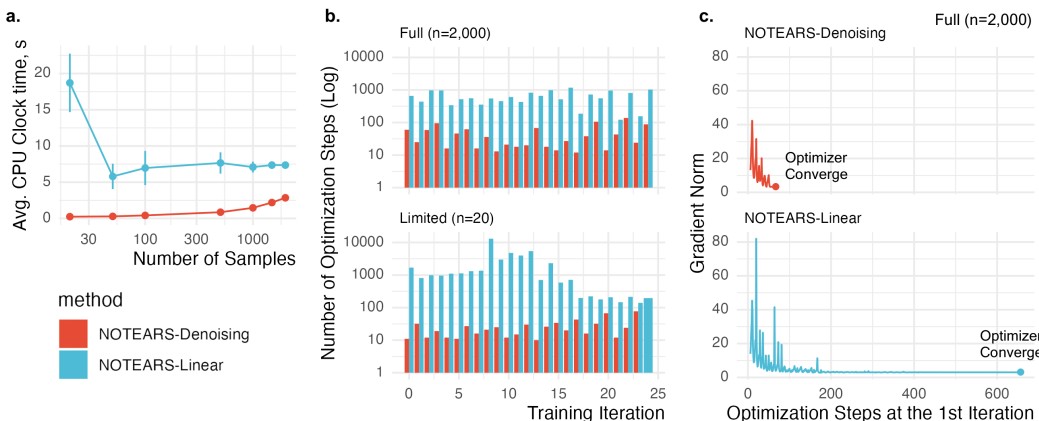

Figure 2: NOTEARS-Denoising runs faster than NOTEARS-Linear because it has smoother gradient. a. Average runtime over 10 runs with various numbers of samples; b. The L-BFGS-B optimizer converges in fewer steps on NOTEARS-Denoising; c. Gradient norm during the first iteration on the Full data (n=2,000). First 5 steps are removed for visualization purposes.

To assess the impact of the choice of k in the k-hop acyclic constraint, we varied the choice of k from 0 to 4 in both SF and ER graphs with 100 nodes under 3 different degrees (10, 20, 30). In all 30 cases, we repeated the experiments 100 times with different ground truth graphs and experiment data.

For real data, we evaluated the proposed methods using the Myocardial Infarction (MI) Complications dataset (Golovenkin et al., 2020) from the UCI data repository, as well as single cell yeast gene expression data (Tjärnberg et al., 2024) (available in GEO with accession number GSE218089 (Edgar R, 2002) ). The MI dataset includes 124 variables and 1,700 observations. We removed the 'ID' column and treated all missing values as 0. Since most categorical variables are leveled with increasing severity, we treated them as continuous variables for simplicity. For the yeast gene expression data, we followed the data preprocessing steps described in the original paper; these details are described in the Appendix. We further removed all ribosomal genes. The final data set has 4,980 genes and 1,428 samples. In both cases, since ground truth is not available, the inferred structures are evaluated using domain knowledge.

### 4.1 Denoising objective leads to smoother gradients

An interesting observation about NOTEARS-Linear (Zheng et al., 2018) is that when the number of samples is very limited, it can take a long time to converge (as shown in Figure 2a). The reason is that the gradient of the L2 loss used in NOTEARS-Linear is in fact not that smooth, so it takes many steps for the L-BFGS-B optimizer to converge in each iteration (as shown in Figure 2b/c). In contrast, with the denoising objective, the gradient is much smoother, so the optimizer only explores a fraction of local minima. A few more comparisons are included in supplement section A.8.

### 4.2 Linear Synthetic Experiment

Figure 3a offers a visual comparison of the inferred networks by DDCD Linear and NOTEARS on a SF graph with 100 nodes. With sufficient numbers of samples (n = 2,000), the inferred network from DDCD Linear is nearly identical to the ground truth network (SHD = 12). In the case where the number of samples is extremely insufficient (n = 20), the main structures of the network can still be visually identified by DDCD Linear while the results from NOTEARS are more limited. Figure 3b provides a broader comparison of the SHD metrics between different methods on different test cases. In general, DDCD Linear, GOLEM, and DAG-GNN (for SF graphs only) are the most competitive methods on these linear cases. Nonlinear methods tend to do worse on these linear cases, which is expected. It is harder to recover ER graphs compared to SF graphs since the signals are weaker and it's more challenging to model on conditional probability.

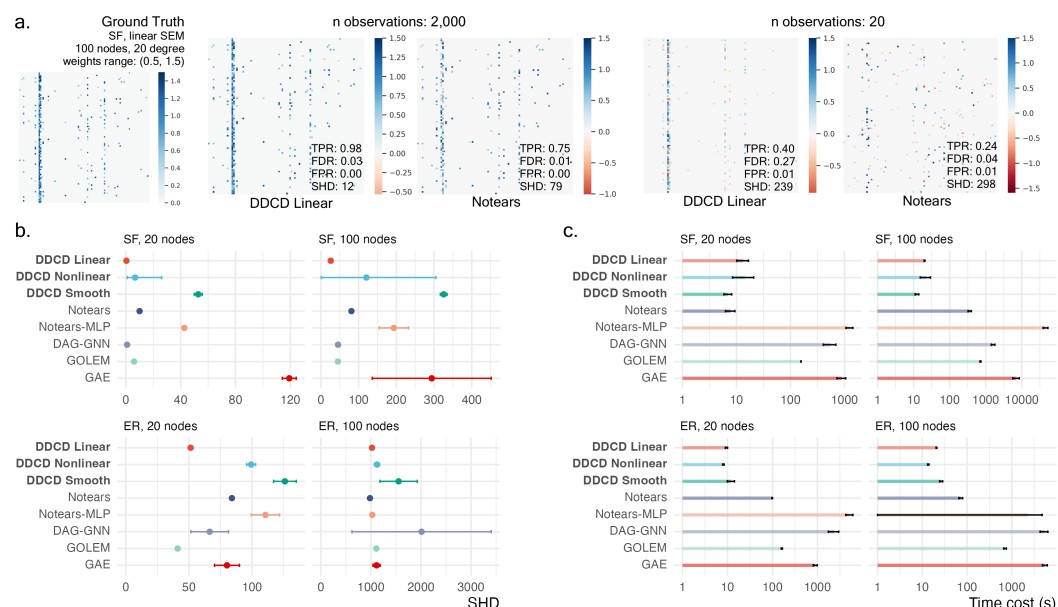

Figure 3: a. Example heatmap of weight estimates on a 100-node scale free graph with different numbers of observed samples using DDCD Linear. b. Benchmark Results on 2,000 observations on linear synthetic data over 10 runs, evaluated using Structural Hamming Distance (SHD). Lower scores indicate better performance. (SF: scale-free; ER: Erdős-Rényi) c. Algorithm execution time on 2,000 observations on CPUs over 10 repeated runs.

In terms of algorithm runtime, all three DDCD models finish execution at a fraction of the cost of other algorithms. The only comparable baseline method is NOTEARS, which finished in an average of 8 seconds and is the fastest algorithm on SF graphs with 20 nodes. However, the time cost quickly scales up to 6 minutes on SF graphs with 100 nodes. In contrast, for DDCDs, the execution time only extends from around 10 seconds to around 20 seconds. For a full comparison of the runtime of DDCDs on a larger network, please refer to supplement section A.4.

### 4.3 NONLINEAR SYNTHETIC EXPERIMENT

On nonlinear benchmark, the DDCD Nonlinear model demonstrates strong performance in recovering the causal structures from nonlinear data. In the example provided in Figure 4a, DDCD Nonlinear not only generates an accurate weight estimate of the graph (TPR: 0.91, SHD: 126), but also provides an approximation to the underlying nonlinear transformation function. After training is complete, we can send the input data $X$ through the trained encoder and decoder and obtain the predicted values for $Y$ and $\hat{X}$. The relationship between $X$ and $Y$ will be the transformation function $f_1$ and the relationship between $YW$ and $\hat{X}$ will be $f_2$. In addition to DDCD Nonlinear, DDCD Linear also presents competitive benchmark performance despite being a linear model.

### 4.4 IMPACT OF K-HOP ACYCLIC CONSTRAINT

In the original NOTEARS constraint, the impact of large circles is reduced by a factorial denominator, as shown in Equation 19 in the supplement. With the k-hop acyclic constraint, we simply ignore those large circles. This raises concerns about whether the inferred graphs are DAGs. To answer this question, we analyzed the number of DAG violations on synthetic graphs with 100 nodes at various degrees. Our results show that small k (as low as 3-hop) is, in fact, good enough to avoid DAG violations in most cases, except for ER graphs with high degrees. The detailed results of the experiment are included in supplement section A.3. Overall, we recommend setting k a little bit higher (e.g. 5 or 10) depending on assumptions about the underlying graphs and computing resources.

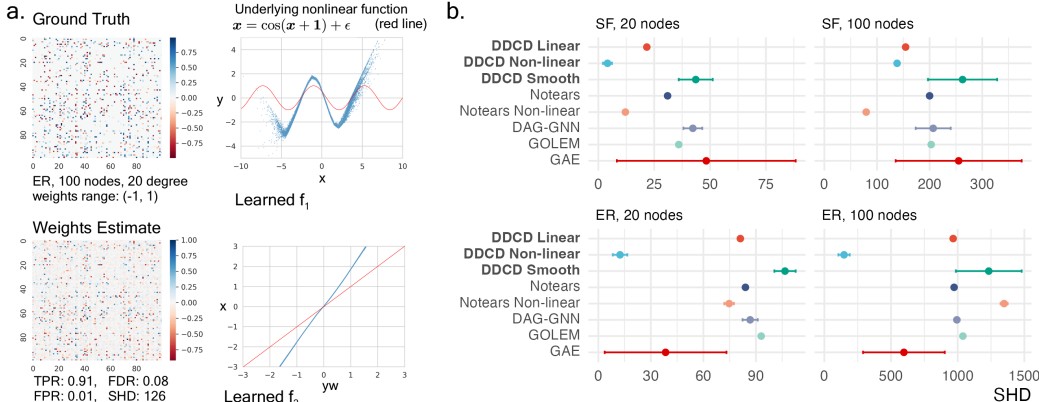

Figure 4: a. Example Weight Estimates on an ER graph with 100 nodes using DDCD Nonlinear. DDCD Nonlinear not only infers the causal structure but also approximate the underlying nonlinear transformation function. (Blue dots: Approximation; Ground truth: red line). b. Benchmark Results with 2,000 observations over 10 runs evaluated using SHD.

### 4.5 REAL WORLD OBSERVATIONAL DATA: MYOCARDIAL INFARCTION

In this section, we assess the performance of DDCD Smooth on the real-world myocardial infarction dataset. After training was complete, we extracted all edges with weights above the cut-off threshold (0.2) in the inferred normalized adjacency matrix and examined a graph of the 2-hop neighbors around the Lethal Outcome (LET_IS) node. As shown in Figure 5, we can identify several meaningful node clusters in this graph, including lethal outcome with its primary causes, critical conditions and interventions, hospital pain control, emergency cardiology pain control, and blood test results, purely based on the topological relationships among the nodes.

The three most important direct causes of lethal outcome include myocardial rupture, cardiogenic shock, and complete Right Bundle Branch Block (RBBB) on ECG at admission; all of these are known to be conditions with a poor prognosis. Cardiogenic shock (K_SH_POST) is further shown to cause "sinus ECG rhythm with heart rate > 90" (ritm_ecg_p_07), consistent with a high heart rate (tachycardia) being a symptom of cardiogenic shock. Pulmonary edema (OTEK_LANC) is shown to cause the use of liquid nitrates in the ICU (NITR_S); this is indeed a common practice for rapidly managing pulmonary edema. Other inferred edges include that NSAID drugs used by the emergency team (NOT_NA_KB) cause blood pressure to increase, and that relapsing pain in the 2nd hospitalization period causes NSAID use in the same period.

There are also some node pairs for which plausible edges are inferred, but in an implausible direction. For example, in the lower right of the figure, "post-infarction angina" (chest pain after the heart attack causing the current hospital admission) is shown to cause "exertional angina pectoris in the anamnesis" (e.g., a reported history of chest pain after exercise), when the former clearly occurs after the latter. Still, many of the directed edges appear consistent with known causal relationships.

### 4.6 REAL WORLD OBSERVATIONAL DATA: GENE REGULATORY NETWORKS

In the yeast gene regulatory network (GRN) analysis with 4,980 genes and 1,428 samples, DDCD smooth required just 34 seconds on GPU, suggesting that the method scales effectively for data sets of this size. It is widely acknowledged that there are many feedback loops in gene regulatory networks (Alm & Arkin, 2003). RegDiffusion, with a denoising architecture related to that of DDCD, is among the fastest and most accurate methods for single-cell-RNA-sequencing-based gene regulatory network inference (Zhu & Slonim, 2024), but it sometimes learns too many cycles. Thus, we compare networks inferred by RegDiffusion (with no DAG constraint), DDCD smooth with a 2-hop DAG constraint, and DDCD smooth with a 10-hop DAG constraint. Surprisingly, we found that requiring acyclicity in the graph, even with just a 2-hop constraint, seems to have a negative impact on the quality of the inferred networks. To illustrate these issues in the inferred networks, we examine the 2-hop neighborhoods around individual genes. Here we show these results for *ADH1*, a key

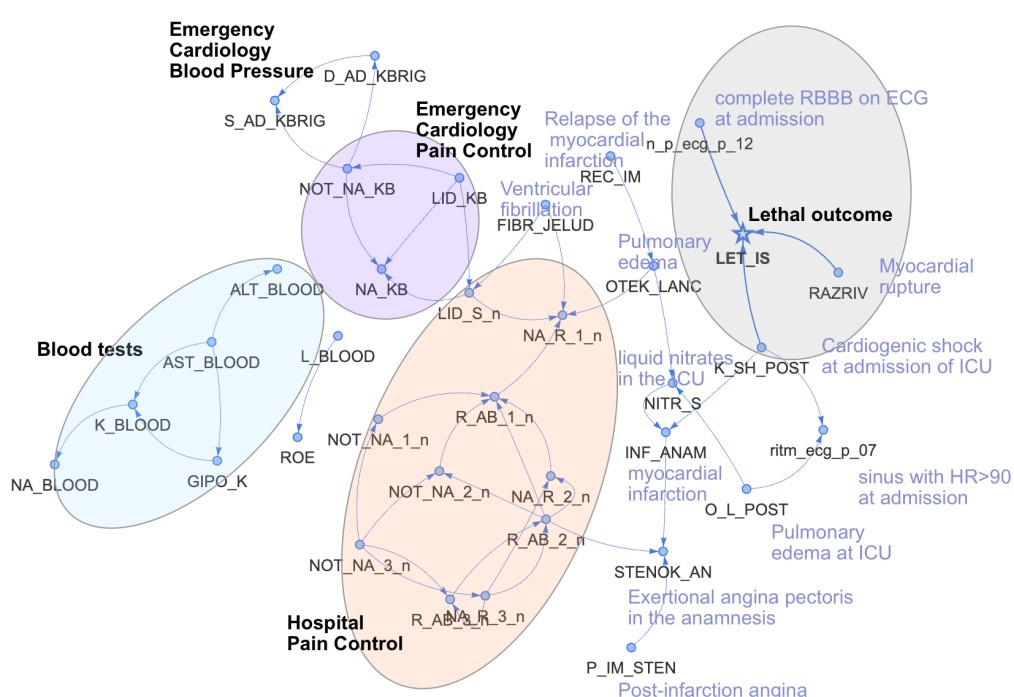

Figure 5: Inferred Causal Network around Lethal Outcome in Myocardial Infarction

player in ethanol fermentation that has been extensively studied in the context of alcohol metabolism and glycolysis (Raj et al., 2014). These results are shown in the Appendix.

## 5  DISCUSSION AND CONCLUSION

Our benchmarks on synthetic data demonstrate the superior performance of the DDCD models in both linear and nonlinear cases. The denoising nature allows the model to explore a broader range of noise and to yield better estimates, especially when the number of observed samples is limited. The capacity for recovering an accurate approximation of nonlinear transformation functions further assists the model in approximating the truth. Since it runs very quickly, it may even be used as a low-cost nonlinearity test in appropriate scenarios.

Compared to existing methods, the most similar model to DDCD is DAG-GNN, which also models the noise term of the SEM equation. To some degree, both DDCD and DAG-GNN train a decoder to reconstruct the original input from pure noise under the constraint of the weighted adjacency matrix. This is similar to the comparison of diffusion models to VAEs with infinite latent spaces (Luo, 2022). By modeling the noise prediction and expected noise under the constraint of the parameterized adjacency matrix at the same time, we eliminate the requirement of doing matrix inversion, which runs in $\mathcal{O}(d^3)$, in DAG-GNN. At the same time, DDCD allows us to experiment with more flexible neural network architectures, addressing multiple types of assumptions.

Since both DDCD Linear and DDCD Nonlinear are based on the same assumptions that NOTEARS, GOLEM, and DAG-GNN make, they also suffer from the same problems pointed out in Reisach et al. (2021) and Kaiser & Sipos (2021). In our real-world experiments, the results from DDCD Smooth are also much more explainable than the results from DDCD Linear and Nonlinear. This once again marks the gap between real and current synthetic data in this field. However, there have been several proposed solutions (Ng et al., 2024; Nazaret et al., 2023) to solve these problem with unequal variance. Incorporating these into our linear and nonlinear models is an important topic for future research.

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

# A APPENDIX

## A.1 EXPLANATION OF THE K-HOP ACYCLICITY CONSTRAINT

In this section, we explain how to derive the proposed k-hop acyclicity constraint in Equation 17 from the NOTEARS DAG constraint in Equation 5.

The NOTEARS DAG constraint is in the form of $h_{\text{NOTEARS}}(\boldsymbol{W}) = \text{tr}(e^{\boldsymbol{W} \circ \boldsymbol{W}}) - d$, where $\circ$ is the Hadamard product, $e^{\boldsymbol{W}}$ is the matrix exponential of $\boldsymbol{W}$, and $\text{tr}()$ is the trace of a matrix. In this case, matrix exponential is the sum of a weighted power series as shown below.

$$e^{\boldsymbol{W}} = \sum_{j=0}^{\infty} \frac{1}{j!} \boldsymbol{W}^j. \tag{18}$$

The trace of the summed matrix and the sum of all the traces are equivalent. At the same time, since $\boldsymbol{W}^0$ is simply the identity matrix, whose trace equals to the value of $d$, we can rewrite the NOTEARS DAG function in the following form:

$$h_{\text{NOTEARS}}(\boldsymbol{W}) = \sum_{j=0}^{\infty} \frac{1}{j!} \text{tr}((\boldsymbol{W} \circ \boldsymbol{W})^j) - d = \sum_{j=1}^{\infty} \frac{1}{j!} \text{tr}((\boldsymbol{W} \circ \boldsymbol{W})^j). \tag{19}$$

In this case, if we want to account for all the cycles within $k$ hops, we can do the following calculation:

$$h_{\text{k-hop}}(\boldsymbol{W}, k) = \sum_{j=1}^{k+1} \frac{1}{j!} \text{tr}((\boldsymbol{W} \circ \boldsymbol{W})^j) \tag{20}$$

As mentioned in the main text, in the case when values in the weighted adjacency matrix are tiny, it might be helpful to multiply the values in the adjacency matrix by a constant multiplier $\gamma$ and then remove it after the trace calculation. Then the equation becomes to the following form:

$$h_{\text{k-hop}}(\boldsymbol{W}, k, \gamma) = \sum_{j=1}^{k+1} \frac{1}{j! \gamma^{2j}} \text{tr}((\gamma \boldsymbol{W} \circ \gamma \boldsymbol{W})^j) \tag{21}$$

If we keep a running product for $j!$, $\gamma^{2j}$, and $(\gamma \boldsymbol{W} \circ \gamma \boldsymbol{W})^j$, we can keep the complexity within $\mathcal{O}(d^2)$.

## A.2 SPECIAL CASE IN DDCD SMOOTH

In DDCD Smooth, all the inputs are transformed into the range of -1 to 1 through MLP and the Tanh activation function. We expected to learn a normalized adjacency matrix $\hat{\boldsymbol{W}}$, where the expected value is $\frac{1}{d}$. This normalized adjacency matrix would be conceptually similar to the normalized adjacency matrix in graph convolution (Kipf & Welling, 2016). Under these assumption, from Theorem we can reach a special form of conclusion that allows us to predict the added noise $\boldsymbol{Z}$ directly.

**Theorem 2.** *With a normalized adjacency matrix, we can directly infer the added noise $\boldsymbol{Z}$.*

*Proof.* Starting from Equation 10, let's pick a random sample $\boldsymbol{x}$ and perturb that with a Gaussian noise vector $\boldsymbol{z} \in \mathcal{N}(0, 1)$ to build $\boldsymbol{x}_t$.

$$\boldsymbol{W}^T \boldsymbol{x}_t = \sqrt{\overline{\alpha_t}} \boldsymbol{W}^T \boldsymbol{x}_0 + \sqrt{1 - \overline{\alpha_t}} \boldsymbol{W}^T \boldsymbol{z}, \tag{22}$$

We can in fact write each element in $\boldsymbol{W}^T \boldsymbol{z}$ as a form of weighted Gaussian mixtures. Since all values in $\boldsymbol{z}$ are standard Gaussian noise with a mean of 0 and variance of 1, the weighted sum of such a mixture will also be centered at 0. Given that the expected value of edge weight in $\boldsymbol{W}$ is $\frac{1}{d-1}$ and there are $d-1$ entries, the expected value for the entire variance is $\sum_{i=0}^{d} \frac{1-\overline{\alpha_t}}{d^2} = \frac{1-\overline{\alpha_t}}{d}$. When $d$ is large, this variance of $\boldsymbol{W}^T \boldsymbol{x}_t$ will be much smaller than the variance term in $\boldsymbol{x}_t$, which is $1 - \overline{\alpha_t}$. When $d$ is really large and the diffusion coeffient is small, we can therefore use $\boldsymbol{W}^T \boldsymbol{x}_t$ to

approximate the unperturbed $x_0$. This argument is very similar to the Central Limit Theorem, but on noise. As a result, $W^T x_t - x_t$ will give us an close estimate of the added noise $z$.

□

### A.3 EXPERIMENT ON THE IMPACT OF k-HOP ACYCLICITY CONSTRAINT

In this experiment, we evaluate the number of DAG violations relative to the choice of k-hop acyclicity in both ER and SF graphs with 100 nodes at 3 different degree levels (10, 20, and 30). In each situation, we generate 100 random graphs and observation data. Figure 6 shows the histogram of the number of DAG violations in those 100 samples under each condition when predicted using DDCD Linear. In most cases, there are no DAG violations for k $\geq$ 3.

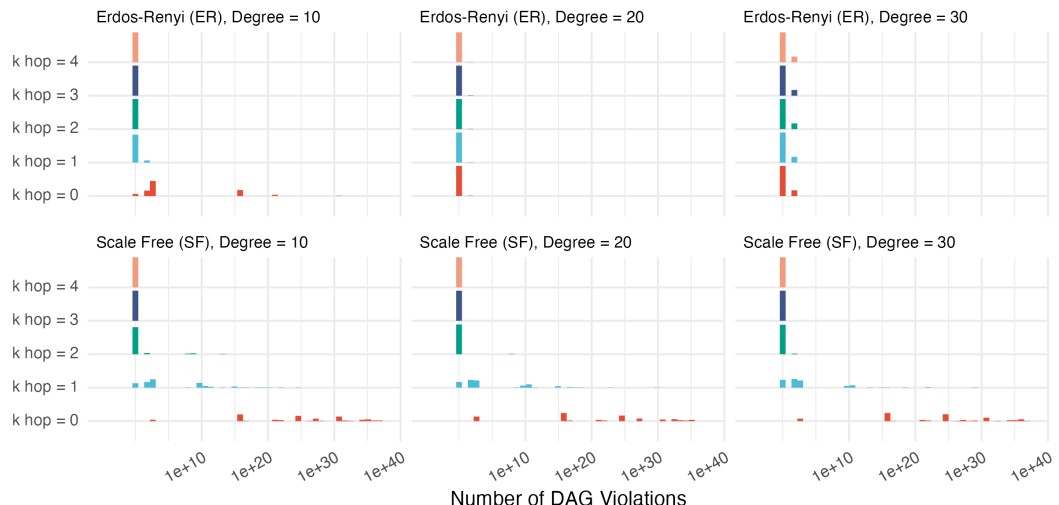

Figure 6: Checking acyclicity within a few hops can effectively avoid DAG violations in most cases.

### A.4 RUNTIME ANALYSIS

Figure 7 shows the execution time of DDCD Linear on SF graphs with different numbers of nodes. The models were executed with different choices of k in acyclicity contraint and on different devices. When graphs are large, GPU acceleration can provide a significant speed gain.

### A.5 PERFORMANCE ON LARGE GRAPHS

Here, we include two sample weight estimates on larger graphs with 1,000 nodes. The main structures of the graphs are recovered (Figure 8).

### A.6 METRICS

Since the inferred graphs are directed graphs, we use the same evaluation methods used in NOTEARS. Since we are not generating non-directed edge predictions at all, here is a simplified description of the metrics that we are using.

1. True Positive Rate (TPR) is defined as

$$\text{TPR} = \frac{\text{True Positive}}{\text{Condition Positive}} \tag{23}$$

   True positive is the number of cases when the predicted association exists in the condition in the correct direction. Condition positive is the total number of true edges in the ground truth graph.

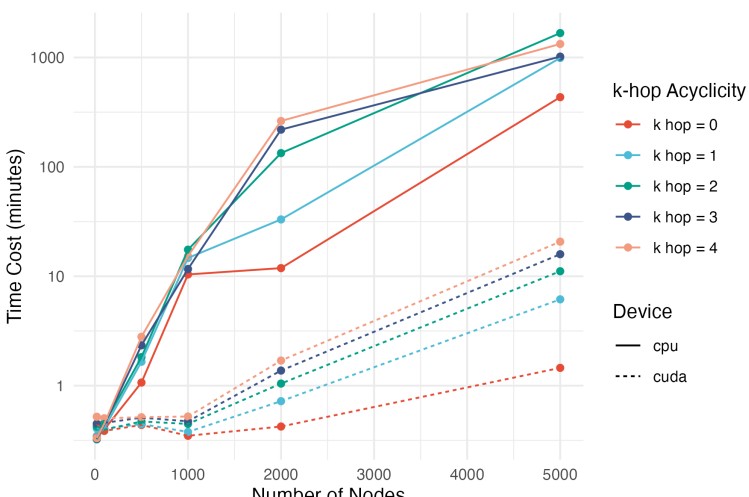

Figure 7: Time cost of running DDCD-Linear on graphs with different sizes on different devices

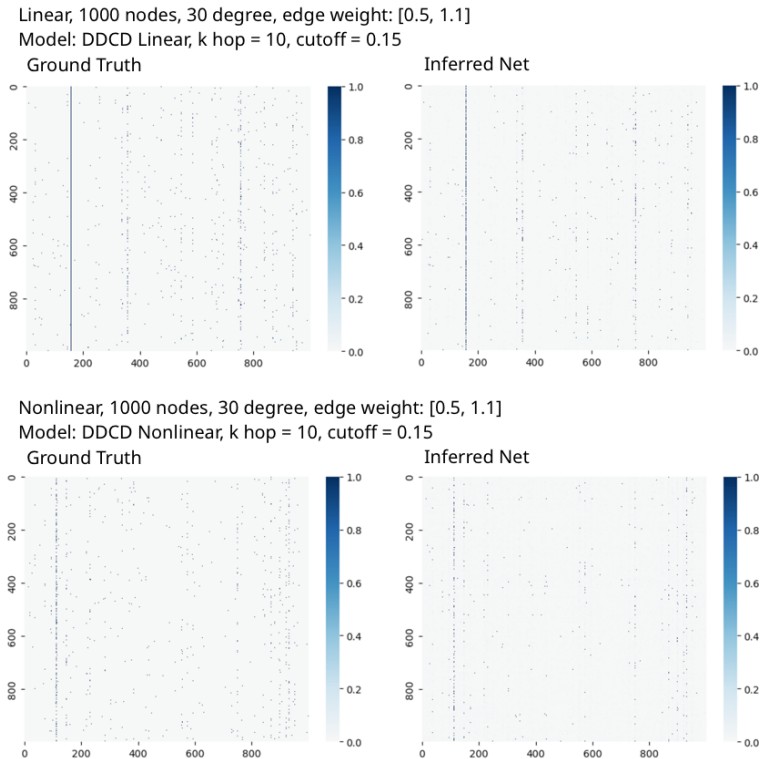

Figure 8: Example Weight Estimates on Graphs with 1,000 nodes. Number of samples in both cases is 2,000.

2. False Discovery Rate (FDR) is defined as

$$\text{FDR} = \frac{\text{False Positive} + \text{Reverse}}{\text{Prediction Positive}} \tag{24}$$

False positive is the number of cases when the predicted association does not exist in the condition. Reverse is the number of cases when the predicted association exists in the condition but in the opposite direction. Prediction Positive is the total number of positive predictions in the inferred graph.

3. False Positive Rate (FPR) is defined as

$$\text{FPR} = \frac{\text{False Positive} + \text{Reverse}}{\text{Condition Negative}} \tag{25}$$

Condition negative is the total number of edges that do not exist.

4. Structural Hamming Distance (SHD) is a measure used to quantify the difference between two directed acyclic graphs (DAGs). It counts the number of operations, including adding an edge, removing an edge, and reversing an edge, required to transform one graph into another. Here, the SHD is the sum of reversed positive predictions, false positive predictions regardless of direction, and false negative predictions regardless of direction.

### A.7 REAL WORLD EXPERIMENTS ON YEAST GENE EXPRESSION

#### A.7.1 DETAILED DATA PREPROCESSING STEPS

We collected the *Saccharomyces cerevisiae* single cell expression data from NCBI's GEO database (Edgar R, 2002) with accession GSE218089 (Tjärnberg et al., 2024). We selected the yeast wild-type strain 2 cultured in nutrient-rich YPD media, as described in Jackson et al. (2020). We followed the standard raw count data pre-processing procedures, which include, (1) gene filtering, by removing genes with positive expression counts in fewer than 10 cells; (2) size factor calculation (Anders & Huber, 2010), which involves calculating the geometric mean of counts across all samples, calculating the ratio for each gene within a sample by dividing each sample's count by the geometric mean for the genes, and calculating the size factor by computing the median of the ratios for each sample; (3) data normalization, normalizing the count by dividing by the size factors; and (4) log transformation, transforming (count-plus-one) using the natural logarithm. After these data processing steps, we performed gene ID conversion using g:Profiler (Raudvere et al., 2019), and we further removed all ribosomal genes (genes with "RPS" or "RPL" prefixes in their names).

#### A.7.2 YEAST GENE EXPRESSION RESULTS

As mentioned in the main text, after data preprocessing, we have 4,980 genes and 1,428 samples. We run RegDiffusion, which does not account for the DAG constraint; DDCD Smooth with a 2-hop DAG constraint; and DDCD Smooth with a 10-hop DAG constraint. All networks are constructed with similar sizes (16 hidden dimensions, 3 layers of MLP blocks) and are trained for 1000 iterations on GPUs. In terms of clock time, RegDiffusion completed in 20 seconds, DDCD Smooth 2-hop DAG in 34 seconds, and DDCD Smooth 10-hop DAG in 109 seconds.

The inferred networks are shown in Figure 9; true positive predictions of neighboring nodes of ADH1 are marked with green circles. Here we treat the STRING protein-protein interaction network (Szklarczyk et al., 2023) as a noisy and non-context-specific ground truth network. In contrast, we would consider the 3 inferred networks to be context-specific since they are inferred from context specific (wild type, cultured in nutrient-rich media) gene expression data.

In the STRING network, *ADH1* is shown to interact with *FBA*, *ENO*, *PDC*, *ADH*, and *SFA*. All three inferred networks successfully captured the links with *FBA*, *ENO*, and *PDC* but their network topologies are quite different.

In the network from RegDiffusion, where we include the top 15 connected candidates for each gene, all three genes are directly connected, forming a chain $(FBA1 \leftrightarrow ENO2) \rightarrow ADH1 \rightarrow PDC1$. DDCD Smooth with the 2-hop constraint forms two pathways starting from *FBA1*: $FBA1 \rightarrow PDC1$ and $FBA1 \rightarrow ADA1 \rightarrow PGK1 \rightarrow ENO2 \rightarrow PDC1$. DDCD Smooth with the 10-hop constraint forms several fragmented pathways, including $PDC1 \rightarrow FBA1$, $PDC1 \rightarrow ADH1$, and $ENO2$, $AHD1 \rightarrow PGK1$.

Although all three models identify proximity to these three neighboring genes, in DDCD Smooth with the 2-hop constraint, neither *ENO2* nor *PDC* is directly connected to *ADH1*. In DDCD Smooth with the 10-hop constraint, neither *FBA1* nor *ENO2* is directly connected to *ADH1*. Further, most of the 2-hop neighboring nodes in the DDCD Smooth (10 hop) plots do not interact with each other, showing that the inferred candidate neighbors lack functional coherence. In contrast, in more accurate inferred networks, the 2-hop neighbors of most genes interact with each other extensively. Based on these and similar observations across the networks, we conclude that the inferred network from

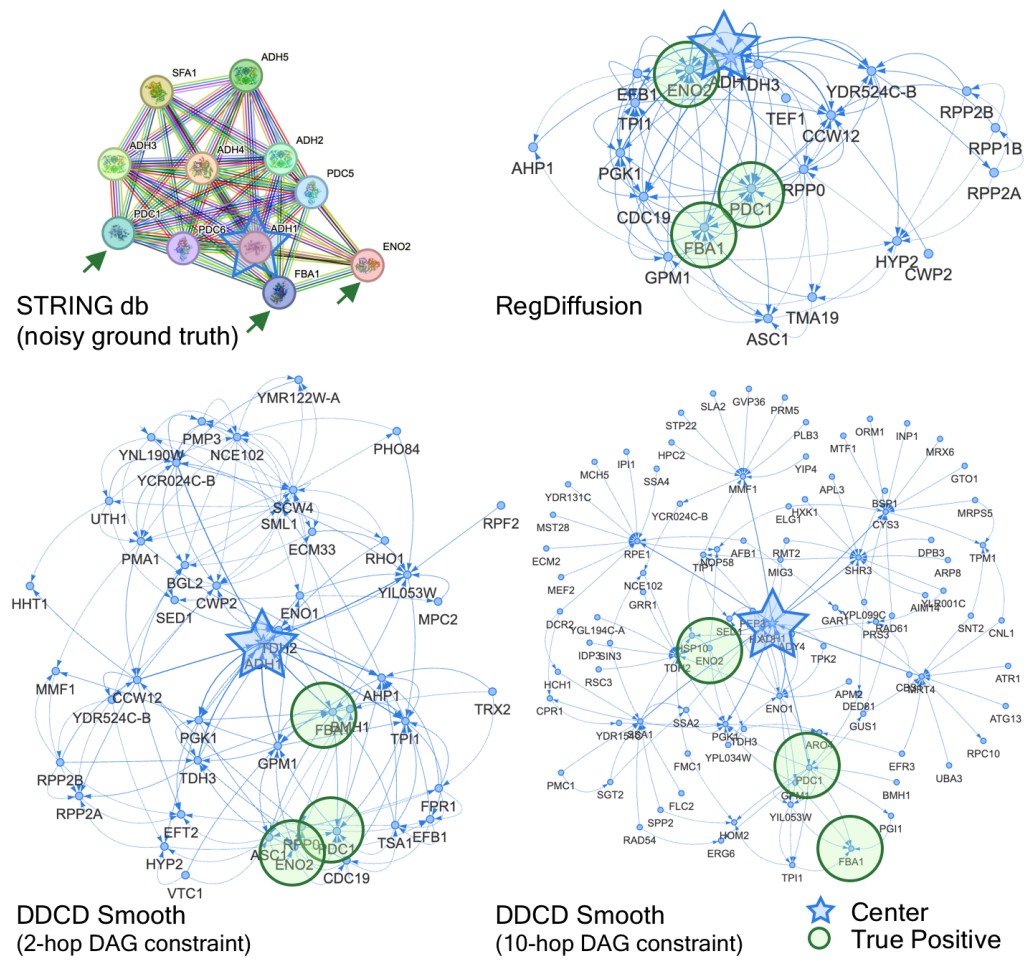

Figure 9: Comparison of the inferred gene regulatory neighborhood around *ADH1* using different models and settings. The network from STRING db is considered as a noisy and non-context-specific ground truth network.

RegDiffusion is more trustworthy. Introducing even limited acyclicity controls to gene networks, or to other networks that potentially include longer feedback loops, may not improve inference results.

## A.8 ADDITIONAL INFERRED EXAMPLES FROM NOTEARS-DENOISING

Here are some additional comparisons between results from NOTEARS-Linear and NOTEARS-Denoising. Overall, as reported in the main paper, when the number of samples is limited (2nd and 4th rows), the results from NOTEARS-Linear (2nd column) may include a lot of noise. This could be resolved by using the denoising diffusion objective (3rd and 4th columns). When the number of samples is sufficient (1st and 3rd rows), in most cases, NOTEARS-Linear will generate very good inference but in some cases such as Sample 2 in row 3 and 4, it may end up in a local minima. On the other hand, using the denoising objective do come with a cost. When the added noise is not small enough, it may introduce some small noisy values in the inferred matrix. This could be resolved by adding smaller noises instead (column 4) but smaller noises will also increase the runtime.

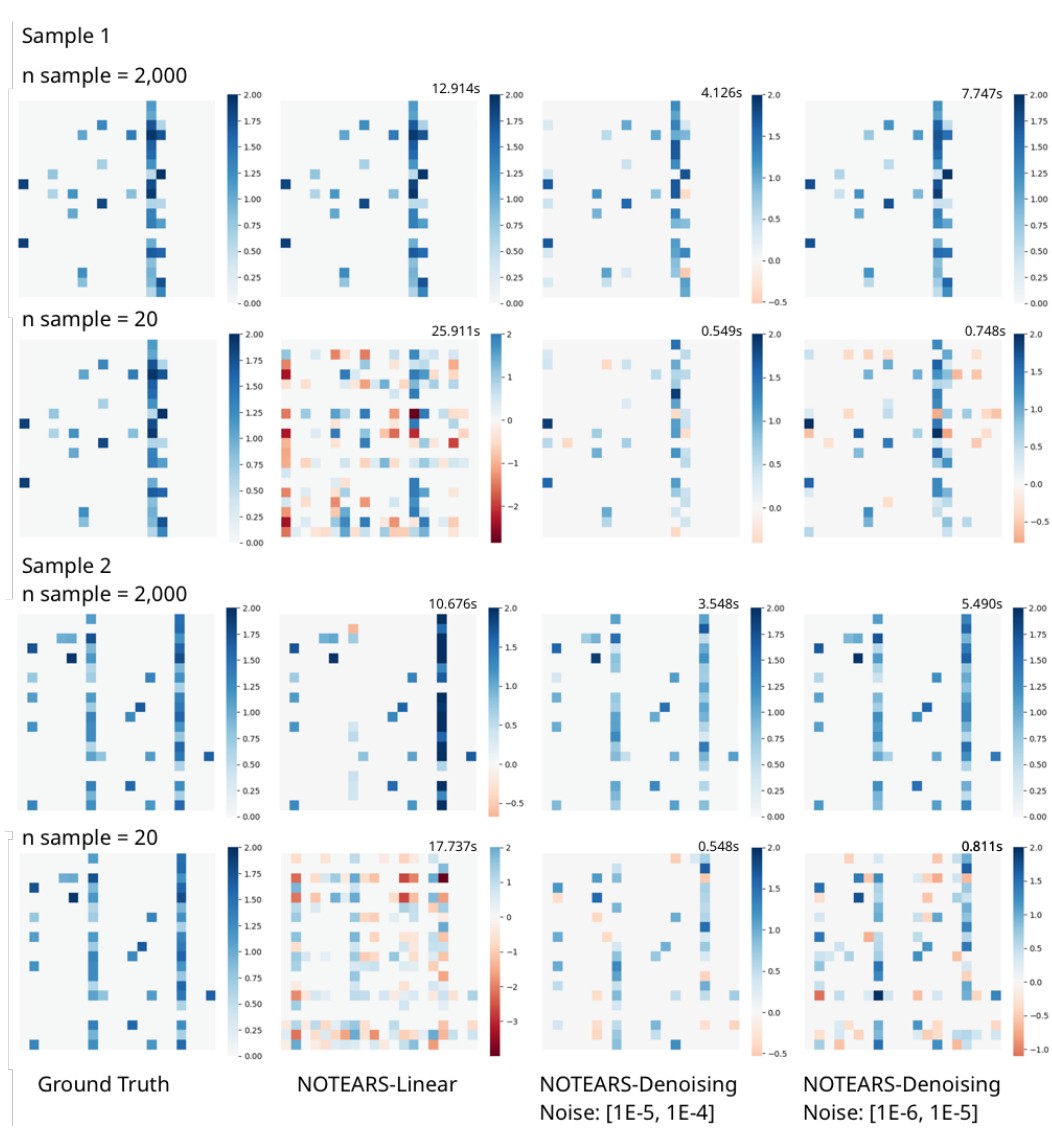

Figure 10: Comparison of some results from NOTEARS-Linear and NOTEARS-Denoising. Execution times are displayed on the top-right corner of each figure.

