# OpenReview forum: "Denoising Diffusion Causal Discovery"
_ICLR.cc/2025/Conference — Submitted to ICLR 2025_

### Official Review · Reviewer_AsnX · 2024-10-24

**Soundness:** 3
**Presentation:** 4
**Contribution:** 4
**Rating:** 6
**Confidence:** 4

**Summary:**

This paper proposes a novel method for causal discovery based on the principles of denoising diffusion. First, it demonstrates that the objective function of a linear Structural Equation Model (SEM) can be transformed into the form of a denoising diffusion model, thereby enabling causal discovery using this methodology. Next, the paper introduces an approach to nonlinear causal discovery by applying the denoising diffusion framework to intermediate variables that are nonlinearly transformed from the original variables. To address the issue of variable scale adjustment, the authors propose a method that smooths the process using the activation function of a Multilayer Perceptron (MLP). Additionally, they incorporate a modified version of the NOTEARS constraint - relaxed to a k-hop condition - into the denoising diffusion process, aiming to better approximate a Directed Acyclic Graph (DAG). Through experiments with synthetic data, the method demonstrates the ability to estimate causal graphs with high accuracy and efficient computation, outperforming traditional causal discovery methods that handle a large number of variables. The approach is further validated with real-world data, showing that it produces results consistent with domain knowledge.

**Strengths:**

The paper successfully presents a natural approach by transforming the objective function of SEM into the form of denoising diffusion. It further extends this concept to nonlinear causal discovery by naturally deriving a model similar to VAE. Moreover, the relaxation of the strict DAG constraint from NOTEARS to a k-hop DAG constraint is smoothly incorporated into the denoising diffusion model. Experimental results using synthetic data demonstrate that the proposed method outperforms traditional approaches in terms of both accuracy and computational efficiency. Additionally, experiments with real-world data further validate the effectiveness of the proposed method.

**Weaknesses:**

This paper is generally well-structured and easy to understand, and the proposed method shows great promise. However, there is room for improvement in the evaluation of the method. The most significant issue lies in the assessment of the DAG property. Since the proposed method introduces a relaxed k-hop DAG constraint rather than enforcing a strict DAG structure, a more careful evaluation is needed to understand how this relaxation impacts the DAG nature of the estimated causal graph. While the Structural Hamming Distance (SHD) between the estimated graph and the ground truth DAG is provided, this metric alone is insufficient to assess how well the DAG constraint is being maintained (or violated). Evaluating the number of loops that occur, for example, could provide more insight. Additionally, allowing loops might enable better handling of feedback loops in real-world data, so evaluating the DAG property in this context is crucial. Including such evaluations could greatly enhance the persuasiveness of the paper.

**Questions:**

Is it possible to provide a quantitative evaluation of how well the DAG constraint is maintained (or violated) in the synthetic data experiments?

---

> ### Author Response · Authors · 2024-11-27
>
> Dear Reviewer,
>
> We truly appreciate your time on reviewing our paper and your support on our work! We are really glad you like it!
>
> Also, thank you for your detailed explanation on how to resolve the concerns on the k-hop acyclicity constraint. Thanks to your suggestion, we designed the new experiments to monitor the number of DAG violations at different k. We believe the new experiment really helped us improve the quality of this work.
>
> The new results are discussed in section 4.4 and supplement section A.3. In most cases (except dense ER graphs), k values as small as 3 can help a 100-node graph avoid any DAG violation. Overall, we recommend people to use slightly higher k based on their assumptions and computing resources.
>
> Thanks again!

---

### Official Review · Reviewer_VBSY · 2024-11-02

**Soundness:** 3
**Presentation:** 3
**Contribution:** 3
**Rating:** 6
**Confidence:** 4

**Summary:**

The study introduces the Denoising Diffusion Causal Discovery (DDCD) framework, which enhances causal structural learning by integrating denoising diffusion probabilistic models (DDPMs). DDCD captures both linear and nonlinear dependencies in causal structures. It employs a k-hop DAG constraint to preserve acyclicity while allowing feedback loops that are common in biological networks. Additional techniques like bootstrap sampling and gradient clipping improve scalability.
The experiments tested DDCD on synthetic (scale-free and Erdős-Rényi) and real-world datasets (myocardial infarction complications and yeast gene networks). DDCD outperformed baseline models (e.g., NOTEARS, DAG-GNN) with lower structural Hamming distance and higher true positive rates, successfully identifying meaningful causal structures in both synthetic and real-world data.

**Strengths:**

The main strength of this paper is its seeming efficiency on large networks due to the fixed-size bootstrap sampling. The DDCD framework captures nonlinear relationships, making it effective for complex real-world applications on both simulated and biological data.

**Weaknesses:**

The inclusion of nonlinear denoising steps increases computational demand which can be exacerbated by the data-intensive need of a diffusion model. The performance could also be affected by fine-tuning needs of the diffusion-related parameters. This may not be trivial in practice. What would you recommend to identify the right hyperparameters? How do you define the threshold values and k-hop constraints to prevent overfitting in densely connected real-world networks?

**Questions:**

The questions stated above:

What would you recommend to identify the right hyperparameters?

How do you define the threshold values and k-hop constraints to prevent overfitting in densely connected real-world networks?

---

> ### Author Response · Authors · 2024-11-26
>
> Dear Reviewer,
>
> Thank you so much for your review and your recommendation on our submission! We appreciate your constructive feedback!
>
> For your questions:
> 1. How to identify the right hyperparameters in nonlinear models? In most cases, we believe a fairly small MLP with residue connection will be able to do the transformation. In our experiment on both nonlinear and smooth models on both synthetic data and real data, 3 linear layers of 16 neurons can give us satisfying results. At the same time, these models tend to be quite insensitive to the size of the network, larger network (deeper or larger with 256 neurons) will likely to generate similar results.
>
> 2. How to determine the thresholding value? Determining the thresholding value is a tricky topic. NOTEARS recommended a cutoff point at 0.3. However, this depended on the fact that in their synthetic data, the weight of the edge was set to be [-2, -0.5] and [0.5, 2]. Having a cutoff point at 0.3 balance out false positive and false negative when the lower bound of the ground truth edge weight is at 0.5. This setup was followed by a lot of following up works and it seems to be a reasonable solution on synthetic data. For real data, it doesn't really make sense to use an very arbitrary value. In our DDCD-smooth model, since we are learning a "normalized" adjacency matrix as described in the paper, all the values in the parameterized adjacency matrix tend to be small. Before we output the predicted matrix, we divide it by the maximum of the absolute value in the matrix such that the maximum or minimum could be normalized to 1 or -1. Then we applied a 0.2 cutoff point as we found that it will help us filter out most of the true positive links. Yet, this number depends on the actual assumptions on the density of the graph. We recommend users to check the distribution of the predicted values before setting up a threshold.
>
> 3. How to choose a good k for k-hop acyclicity? This time, to resolve some concerns on whether limited k-hop acyclicity contraint will introduce DAG violations, we added an experiment and counted the number of DAG violations under different choice of k. Our experiment, as discussed in section 4.4 and supplement section A.3, showed that unless the graph is Erdos-Renyi with high degree of edges, even small k (e.g. k = 3) can reduce the number of DAG violations to 0. At the same time, we would like to argue that in NOTEARS' DAG constraint, the factorial denominator in matrix exponential could effectively diminish the impact of large circles as well. Overall, we recommend users to use a high-enough k for acyclicity control based on their assumptions and computing resource. A good default for k would be 5 or 10. In fact, I would consider k=10 being very high since $\frac{1}{10!} = \frac{1}{3628800}$ and that is the impact of one 10 hop cycle on the NOTEARS constraint.
>
> I hope that answers all your questions! Thanks again for your positive review! Feel free to let us know if you have any other questions.

---

> > ### Comment · Reviewer_VBSY · 2024-12-03
> > **Thanks**
> >
> > Thank you for your answer. This is very informative. I would suggest restructuring the text of your manuscript to include a discussion on the network size, thresholding value, and choosing k for the k-hop.

---

> > > ### Author Response · Authors · 2024-12-03
> > >
> > > Sounds good. We will add that in a future version of this paper. Thanks for your suggestion! =)

---

### Official Review · Reviewer_yd3d · 2024-11-04

**Soundness:** 2
**Presentation:** 3
**Contribution:** 2
**Rating:** 3
**Confidence:** 5

**Summary:**

This work proposes a new Denoising Diffusion Causal Discovery (DDCD) framework that leverages Denoising Diffusion Probabilistic Models for causal discovery under additive noise setting.

**Strengths:**

- This work proposes a denoising optimization as effective as Notears [1].

**Weaknesses:**

- There is also some similar work that used the diffusion model to learn causality under ANM [2]. What is the difference compared with it?
- This work seems only applicable in the Gaussian noise setting, can it be extended to the non-Guassian setting?
- Why the diffusion can help improve the causal discovery compared with the other Notears-based method. Can you offer some theoretical guarantee for it?
- Besides, whether will the nonlinear loss function (16) identify the true causal structure since it is just a combination of the linear loss of $Y$ and the nonlinear loss of $X$ without theoretical guarantee.
- The proposed "k-hop DAG constraint" aiming to replace the acyclic constraint seems to lack theoretical analysis and the result of real-world experiments does not appear to outperform the Regdiffusion.
- In lines 39-40, PC and LiNGAM are not Score-based approaches.

[1]Zheng, Xun, et al. "DAGs with NO TEARS: Continuous Optimization for Structure Learning." Neural Information Processing Systems,Neural Information Processing Systems, Jan. 2018.
[2]Sanchez, Pedro, et al. "Diffusion Models for Causal Discovery via Topological Ordering." The Eleventh International Conference on Learning Representations. 2023.

**Questions:**

See the weakness above.

---

> ### Author Response · Authors · 2024-11-26
>
> Thank you so much for your thoughtful comments! We realized that we lacked some clear explanation and evidence on some core issues in the previous version of this manuscript. We have added experiments and revised our manuscripts based on your comments.
>
> For your questions:
> 1. Yes, DiffAN is another diffusion model based causal discovery model with a very impressive design. However, DDCD and DiffAN are fundamentally different. In DDCD, we focus on directly learning the adjacency matrix as part of the model parameter. DiffAN, on the other hand, focuses on iteratively finding leaf node and learning the graph via topological ordering. Also, DiffAN does not require a DAG constraint, which sets it apart from most of the acyclicity score based methods. Either way, we added DiffAN to the review part of the introduction.
>
> 2. We would love to run more experiments with non-Gaussian setting but unfortunately this time we don't have enough time to run a systematic evaluation. We are happy to include the results in the supplement in the future version of this manuscript.
>
> 3. We realized we did not provide clear evidence on how the denoising diffusion process could help learning. Therefore, we implemented a toy model called NOTEARS-Denoising with the denoising diffusion process, while everything else is the same as the original implementation of NOTEARS-Linear. With this toy model, we demonstrate that the denoising diffusion process smoothes out the gradient norm and helps the model to avoid large updates on its parameters and converge must faster. We think it also allows a flatter optimal point and increase the robustness when the number of samples is limited.
>
> 4. Thanks for pointing out the unclarity. We have updated the writing in Section 3.2. This time we made it clear that by assuming Y=YW, we are basically creating an autoencoder on the input data X while keeping all the dimensions in the latent mapping variable Y. f1 is the encoder and f2 is the decoder. Since all the dimensions are preserved, if we can identify an causal relationship W on the latent variable Y, this relationship is very likely to preserve for variables in X. To learn W in Y, we can use the linear denoising trick. We also provided empirical evidence on the successfully learned non-linear transformation function. I hope it's convincing enough.
>
> 5. We did provide a theoretical analysis of NOTEARS's constraint and our k-hop constraint in supplement section A1.  Basically, in NOTEARS' DAG constraint, the factorial denominator in matrix exponential could effectively diminish the impact of large circles while in our case, we simply set them to be 0. To further resolve the concern on DAG violations, we added an experiment and counted the number of DAG violations under different choice of k. Our experiment, as discussed in section 4.4 and supplement section A.3, showed that unless the graph is Erdos-Renyi with high degree of edges, even small k can reduce the number of DAG violations to 0. At the same time, we would like to argue that in NOTEARS' DAG constraint, the factorial denominator in matrix exponential could effectively diminish the impact of large circles as well.
>
> 6. Thanks for pointing out that mistake. We have completely rewritten that section with correct description.
>
> Again, thanks for your suggestions, which really helped us improve this manuscript. Feel free to let us know if there are any other standing questions.

---

### Official Review · Reviewer_npBw · 2024-11-04

**Soundness:** 2
**Presentation:** 3
**Contribution:** 2
**Rating:** 3
**Confidence:** 4

**Summary:**

This paper studies the causal discovery problem by reformulating it into a denoising diffusion process problem. It proposes three frameworks, the linear model, the nonlinear model, and the smooth model. Under the linear model, the authors show the equivalence of the denoising objective to the objective of the structural equation model. Furthermore, the author proposes the k-hop DAG constraints to enforce DAG structures within k-hops of the nodes. This method speeds up the computation by only checking the DAG of local neighbors. The paper compares the proposed method with several baselines including DAGs and the proposed method shows improvement in SHD score.

**Strengths:**

The scalability of the directed graph discovery procedure is an important and yet challenging problem. This paper addresses the issue by proposing a k-hopDAG constraint, which greatly reduces the computation time.

The simulation results indicate that the proposed method works well under the nonlinear structure.

**Weaknesses:**

It is unclear why formulating the causal discovery problem as a diffusion process is needed and how it can make the estimation scalable. It seems that the major trick that can make the estimation procedure scalable is the k-hop DAG constraint, which reduces the algorithm's runtime. However, under the k-hop framework, the DAG structure is no longer guaranteed. It is not clear if a graph with cycles can be uniquely identified under this framework.

**Questions:**

a. It would be great to add more recent baselines, e.g., Bello et al., 2022; Deng et al., 2023, to the simulations.

b. It is not clear why Equation (13)-(14) is equivalent to Equation (4), as $Y$ has an additional term $E_1$.

c. In the experiment section, it states that the number of nodes ranging from 20 to 5000 are tested, but only results of 20 and 100 are reported. How does the method perform when the number of nodes increases beyond 100?'

d. The first term of  Equation (16) is
$\|(X−f_2(f_1(X)W))+(Y_t−Y_tW)−diag(\sqrt{1−\bar\alpha_t})Z(I−W)\|_F^2$. I am a bit confused here. Why is it not the decomposition of two terms $\|(X−f_2(f_1(X)W))\|_F^2+\|(Y_t−Y_tW)−diag(\sqrt{1−\bar\alpha_t})Z(I−W)\|_F^2$?

---

> ### Author Response · Authors · 2024-11-26
>
> Dear Reviewer,
>
> We appreciate the thoughtful comments and suggestions! Thanks to your review, we realized that our previous submission lacked some convincing evidence on some core issues and therefore caused some misunderstanding. We have added experiments and revised our manuscript.
>
> weakness 1. We realized we did not provide clear evidence on how the denoising diffusion process could help learning. Therefore, we implemented a toy model called NOTEARS-Denoising with the denoising diffusion process, while everything else is the same as the original implementation of NOTEARS-Linear. With this toy model, we demonstrate that the denoising diffusion process smoothes out the gradient norm and helps the model to avoid large updates on its parameters and converge must faster. We think it also allows a flatter optimal point and increase the robustness when the number of samples is limited.
>
> weakness 2. One of the concerns that you mentioned is that under the k-hop framework, the DAG structure is no longer guaranteed. This concern is certainly valid. To answer this question, we counted the number of DAG violations under different choice of k. Our experiment, as discussed in section 4.4 and supplement section A.3, showed that unless the graph is Erdos-Renyi with high degree of edges, even small k can reduce the number of DAG violations to 0. At the same time, we would like to argue that in NOTEARS' DAG constraint, the factorial denominator in matrix exponential could effectively diminish the impact of large circles as well.
>
> Question a. The work from Bello et al., 2022; Deng et al., 2023 are included in our background review. Both are indeed amazing DAG learning tools with very impressive performance. However, we don't have enough time this time to carry out a systematic comparison with these two methods. We are happy to include these comparison in a future version of this paper.
>
> Question b. Thanks for pointing out the error! The error term in Eq 13 is not necessary at all. We have removed it from the equation.
>
> Question c. Sure, we have added a comparison of runtime on larger graphs (20-5,000 nodes) in Supplement section A.4. We also added two sample weight estimates (both linear and nonlinear) on graphs with 1,000 nodes in Supplement section A.5
>
> Question d. Thanks for pointing it out! We have fixed the notation.
>
> Thanks in advance for taking your time to review out updates! Please feel free to let us know if you have any additional questions or comments!

---

> > ### Comment · Reviewer_npBw · 2024-11-26
> >
> > Thank you for addressing my comments and providing additional experiments.
> > 1. Comparison of NOTEARS-denoising and NOTEARS-linear. Can you also compare the quality of the graph recovery for two methods? My concern is that even if it converges faster, it might converge to bad local optima. In addition, for Figure 2-(c), it does not appear to me that NOTEARS-denoising converges significantly faster.
> >
> > 2. Can you further explain why the diffusion model would have a smoother gradient? Can we infer that from the equation?
> >
> > 3. Thanks for running the additional experiment to address my concern. I worry about the rigorousness of the claim. It is because (1) according to Figure 6, the x-axis has a very large scale, it seems to me that the number of violations is large. It is not clear to me how generalizable the algorithm is. (2) I buy the claim that k-hop might be a potential alternative with lossy output to scale up the graph recovery problem. However, there is no theoretical guarantee in the worst-case scenario. If it is possible to provide an upper bound on some cases, in my opinion, this would be a much more solid paper.

---

> ### Author Response · Authors · 2024-11-27
>
> Thank you so much for getting back to us so quickly!
>
> For your questions:
> 1. We just added some comparisons in supplement section A8. Overall, for this toy model, I would say the results are quite comparable. In most cases, if the added noise is too much, it may degrade the final output but that could be resolved by introducing smaller noises while still enjoying shorter converge time. At the same time, when the number of samples is not sufficient, NOTEARS-Linear in fact tend to converge to bad local optima while the results from NOTEARS-Denoising seem to be much closer. We even find out some cases (~10-20 %) that NOTEARS-Linear landed on bad local optima while NOTEARS-Denoising successfully avoids those cases. For your comment on improvements on runtime, in Figure 2c, in the 1st iteration of optimization, NOTEARS-Denoising converged within ~70 steps while NOTEARS-Linear took almost 700 steps. That's almost 10x improvement on converge speed.
>
> 2. As we mentioned in Section 3.1, introducing noise and capture the noise with the new objective helps find solutions that generalize better by avoiding sharp minima. Here I'm quoting from a 2016 paper on deep learning optimization and generalization "it appears that noise in the gradient pushes the iterates out of the basin of attraction of sharp minimizers and encourages movement towards a flatter minimizer where noise will not cause exit from that basin"[1]. I believe this is at least part of the reasons. Right now it's a little difficult to directly infer that from the equation. Part of the reason is that we can't really explain why NOTEARS-Linear has those sharp updates on its gradient norm as you can see in Figure 2c.
>
> 3. We are sorry if the scales of Figure 6 might be misleading. The large x-axis scales in Figure 6 are there for cases when k is extremely small (eg. k = 0, 1, or 2). As we mentioned in the paper, the number of DAG violations often went back to 0 in most cases except for densely connected ER graphs, which is likely the worst-case scenario. We agree with you that more theoretical exploration on this is definitely necessary for us to understand the behavior of k-hop DAG but that might go beyond the scope of this paper (we are at 10 page sharp). We are happy to do more exploration in a future project.
>
> Thanks again for your quick response and we are looking forward to hearing from you what you think.
>
> [1] Keskar, Nitish Shirish, et al. "On large-batch training for deep learning: Generalization gap and sharp minima." arXiv preprint arXiv:1609.04836 (2016).

---

### Meta-Review · Area_Chair_TKon · 2024-12-20

**Metareview:**

This paper proposes a Denoising Diffusion Causal Discovery (DDCD) framework that reformulates the causal discovery problem into a denoising diffusion process. The paper is novel in combining the denoising diffusion for causal discovery. However, as raised by the reviewers, the motivation and the theoretical analysis are somewhat limited.

**Additional Comments On Reviewer Discussion:**

Although the paper has some merits, such as the scalability of the directed graph discovery procedure and the ability to capture both linear and nonlinear dependencies, the issues raised by the reviews are critical. For instance, the lack of clarity on why formulating the problem as a diffusion process is necessary and how it contributes to scalability (npBw), the absence of theoretical guarantees for the proposed nonlinear loss function and the k-hop DAG constraint (yd3d), and the need for better evaluation of the DAG property under the relaxed k-hop constraint (AsnX). The paper still needs a major revision before it can be accepted.

---

### Decision · Program_Chairs · 2025-01-22

Reject